# Generalization Error Analysis of Quantized Compressive Learning

**Xiaoyun Li**
Department of Statistics
Rutgers University
Piscataway, NJ 08854, USA
`xiaoyun.li@rutgers.edu`

**Ping Li**
Cognitive Computing Lab
Baidu Research
Bellevue, WA 98004, USA
`liping11@baidu.com`

## Abstract

Compressive[1] learning is an effective method to deal with very high dimensional datasets by applying learning algorithms in a randomly projected lower dimensional space. In this paper, we consider the learning problem where the projected data is further compressed by scalar quantization, which is called quantized compressive learning. Generalization error bounds are derived for three models: nearest neighbor (NN) classifier, linear classifier and least squares regression. Besides studying finite sample setting, our asymptotic analysis shows that the inner product estimators have deep connection with NN and linear classification problem through the variance of their debiased counterparts. By analyzing the extra error term brought by quantization, our results provide useful implications to the choice of quantizers in applications involving different learning tasks. Empirical study is also conducted to validate our theoretical findings.

## 1 Introduction

Random projection (RP) method [36] has become a popular tool for dimensionality reduction in many machine learning and database applications, e.g., [14, 2, 11, 4, 38, 9], including classification, matrix sketching, compressive sensing, regression, bioinformatics, matrix factorization, etc. The great success of random projection lies in the favorable distance preserving property with fairly elegant statement given by the famous Johnson-Lindenstrauss Lemma [19, 10]. In short, under some conditions we can always project a set of $n$ points $X \in \mathbb{R}^{n \times d}$ in a high-dimensional space onto a lower $k$-dimensional space such that pair-wise distances are approximately preserved, with high probability. Here $k \ll d$ is the number of random projections. This nice theoretical guarantee has originated the study of generalization performance of learning in the reduced dimensional space instead of the original space. This line of work is called *compressive learning* [16, 3, 30, 13, 20, 34, 35, 21].

In many cases, it is useful to further condense the projected data, due to storage saving, privacy consideration, etc. Consequently, research on quantized random projections (QRP) has been conducted for a while, i.e., [24, 25]. QRP itself has been developed into many promising fields in computer science, such as 1-bit compressive sensing, simhash and so on [32, 1, 6, 23]. More recently, it is shown that quantized random projection is also very convenient for cosine estimation and similarity search [27, 26, 28]. However, to the best of our knowledge, theoretical analysis of QRP in learning mechanisms is still missing in literature. In this paper, we investigate the generalization error bounds of applying QRP in three models: nearest neighbor classifier, linear classifier and least squares regression. Apart from finite $k$ analysis, we also consider the case where $k$ is asymptotically large.

**Contributions.** An important implication of our analysis is to answer the following question—The generalization performance using quantized random projections is determined by what factors of a quantizer? Our theoretical analysis illustrates that for nearest neighbor and linear classification, the

extra loss of quantization decreases as $k$ gets large, and the learning performance is determined by the variance of debiased inner product estimator when data samples are allocated on the unit sphere. For regression problems, the distortion of a quantizer becomes crucial. Our theoretical findings are validated by empirical study. Practically, our results also suggest appropriate quantizing strategies for different learning models, which would be helpful for various applications.

## 2 Preliminaries

**Problem setting.** Assume dataset $X, Y \sim \mathcal{D}^n$ with $X = [x_1, ..., x_n]^T \in \mathbb{R}^{n \times d}$, and $x_i, i = 1, ..., n$ are $i.i.d.$ drawn from some marginal distribution $\mathcal{X}$. Throughout this paper, we assume that every sample in $X$ is standardized to have unit Euclidean norm[2]. Therefore, the domain of $\mathcal{X}$ is the unit Euclidean sphere $\mathcal{S}_d$, which allows us to call "inner product" and "cosine" interchangeably. For classification problems, $Y \in [0, 1]^n$, while in regression model $Y \in \mathbb{R}^n$. We will focus on the Gaussian random projection matrix $R = [r_1, ..., r_k] \in \mathbb{R}^{d \times k}$ with $i.i.d.$ standard normal entries. Random projection is realized by $X_R = \frac{1}{\sqrt{k}} X R$, where the factor $\frac{1}{\sqrt{k}}$ is for the ease of presentation.

**Quantized RP's.** An $M$-level scalar quantizer $Q(\cdot) : \mathcal{A} \to \mathcal{C}$ is specified by $M + 1$ decision borders $t_0 < t_1 < \cdots < t_M$ and $M$ reconstruction levels (or codes) $\mu_i, i = 1, ..., M$. Given a signal $v$, the quantizing operator is defined as $Q_b(v) = \mu_i \in \mathcal{C}$, such that $t_{i-1} < v \le t_i$. Here, $\mathcal{A}$ is the domain of the original signal and $\mathcal{C}$ is the set of codes. The number of bits is defined as $b = \log_2 M \ge 1$. We note that $t_0$ and $t_M$ can be either finite or infinite depending on the support of signal. For generality, in this paper we do not restrict our analysis to any specific quantizer, but cast basic assumption of increasing and bounded codes, i.e., $\mu_1 < \cdots < \mu_M$ and $t_{i-1} < \mu_i < t_i$ for all $i = 1, ..., M$.

**Definition 1.** *(Maximal gap) For an $M$-level quantizer $Q$ defined above and an interval $[a, b]$, denote $\alpha = \{i : t_{i-1} < a \le t_i\}$ and $\beta = \{i : t_i < b \le t_{i+1}\}$. The **maximal gap** on a interval $[a, b]$ is defined as the largest distance between any two nearby borders in $[a, b]$, $g_Q(a, b) = \max\{\max\limits_{i: \alpha \le i \le \beta - 1} |t_{i+1} - t_i|, |t_\alpha - a|, |b - t_\beta|\}$, if $t_\alpha \in [a, b]$, and $g_Q(a, b) = |b - a|$ otherwise.*

In a random signal model, $v$ is assumed to be generated from a probability density $V \sim f$. In this case, the following is an important quantity measuring the information loss of a quantizer.

**Definition 2.** *(Distortion) The distortion of a $b$-bit quantizer $Q_b$ with respect to distribution $f$ is*

$$D_b = E_{V \sim f}[(V - Q_b(v))^2] = \int (v - Q_b(v))^2 f(v) dv. \tag{1}$$

*Uniform quantizer* is the most simple quantizer, whose partitions are equal size bins with length $\triangle$ (i.e., $t_{i+1} - t_i = \triangle, \forall i$ with finite $t_i, t_{i+1}$) and the reconstruction levels are simply the mid points of the bins. *Lloyd-Max (LM) quantizer* [29, 31] is designed to minimize the distortion with respect to a given distribution. In this present paper, we optimize LM quantizer with respect to standard normal distribution, since any $r_i^T x$ with $i = 1, .., k, x \in \mathcal{X}$ is marginally $N(0, 1)$ under Gaussian RP's. Now suppose $Q$ is a quantizing function that operates element-wise on matrix. The quantized RP is defined as $X_Q = \frac{1}{\sqrt{k}} Q(XR)$. We are interested in using $X_Q$ for learning problems instead of $X$.

**The inner product estimate.** It is easy to show that for $x_1, x_2 \sim \mathcal{X}$ with $\rho_{12} = \cos(x_1, x_2)$, the projections $(R^T x_1, R^T x_2)$ consist of $k$ $i.i.d.$ samples from $N\left( \begin{pmatrix} 0 \\ 0 \end{pmatrix}, \begin{pmatrix} 1 & \rho \\ \rho & 1 \end{pmatrix} \right)$. One important application is to use the projections to estimate $\rho_{12}$. It is well-known that the inner product of two projected vectors is an unbiased estimator of $\rho_{12}$, i.e., $E[\hat{\rho}_R] = E[\frac{x_1^T R R^T x_2}{k}] = \rho_{12}$. This estimator is called the full-precision estimator. For quantized RP's, we analogously define the quantized estimator as $\hat{\rho}_Q = \frac{Q(R^T x_1)^T Q(R^T x_2)}{k}$, whose statistical property is studied in [27, 28]. In most cases, $\hat{\rho}_Q$ is biased. The following analytical concept is considered in [28], which is also helpful in our analysis.

**Definition 3.** *(Debiased variance) Denote the space of expectation of estimator $\hat{\rho}_Q$ as $\mathcal{E}$. If there exists a map $g : [-1, 1] \to \mathcal{E}$, the **debiased estimator** is defined by applying the inverse map $\hat{\rho}_Q^{db} = g^{-1}(\hat{\rho})$ to correct for the bias. The variance of $\hat{\rho}_Q^{db}$ is called the **debiased variance**.*

# 3 Quantized Compressive Nearest Neighbor Classification

We first look at the generalization error incurred by learning using $X_Q$ instead of $X$ on nearest neighbor (NN) classification problem, which is a simple but powerful non-parametric algorithm that is popular in practice. Given a dataset $S = (X, Y)$ and a test sample $(x, y) \sim \mathcal{D}$ where $y$ is unknown, the algorithm finds the nearest neighbors of $x$ in $X$, denoted by $(x^{(1)}, y^{(1)})$, and classifies $x$ as $\hat{y} = y^{(1)}$. We denote the classifier of NN as $h_S(x) = y^{(1)}$, in the original sample space. Denote the conditional distribution of $y$ given $x \sim \mathcal{X}$ as $\eta(x) = P(y = 1|x)$. A Bayes classifier, $h^*(x) = \mathbb{1}\{\eta(x) > 1/2\}$, is well known as the optimal solution in minimizing the risk $\mathcal{L}(h(x)) = E_x[\mathbb{1}\{h(x) \neq y\}]$ over all hypothesis. [8] showed that the risk of NN classifier converges to $2\mathcal{L}(h^*(x))$ as sample size $n \to \infty$. See additional asymptotic analysis in [15, 37, 18]. In finite $n$ case, [33, 17, 7] studied the error bounds and convergence rate of NN classifier, all of which require the sample size $n$ increases exponentially in dimensionality $d$, under some Lipschitz-type assumptions on the conditional probability function $\eta(x)$. As discussed in [33, 21], by the celebrated No-Free-Lunch Theorem [39], this exponential sample complexity comes from the nature of this problem and cannot be reduced in general.

**Classical finite sample analysis.** Yet, the work [21] demonstrates that when data has small "metric size" measured by metric entropy integral $\gamma$ (which will be defined later), it is possible to reduce the sample complexity from $O(e^d)$ to $O(e^\gamma)$ by working in the randomly projected space using $X_R$. This is called compressive NN classification. The following definitions are necessary for our analysis.

**Definition 4.** *Let $(\mathcal{T}, \|\cdot\|)$ be a totally bounded metric space, and $\alpha > 0$. $\mathcal{T}$ is $\alpha$-separated if $\forall a - b \in \mathcal{T}, a \neq b, \|a - b\| \geq \alpha$ holds. The $\alpha$-packing number of $\mathcal{T}$ is $N_{\|\cdot\|}(\alpha, \mathcal{T}) = \max\{|\mathcal{T}'| : \mathcal{T}' \text{ is } \alpha\text{-separable}, \mathcal{T}' \subset \mathcal{T}\}$.*

**Definition 5.** *The $\alpha$-entropy of $\mathcal{T}$ is defined as $Z(\alpha, \mathcal{T}) = \log N(\alpha, \mathcal{T})$, and function $Z(\cdot, \mathcal{T})$ is called the metric entropy of $\mathcal{T}$.*

**Theorem 1.** *[22]. Let $\mathcal{X} \subset \mathbb{R}^d$, and $R \in \mathbb{R}^{d \times k}$ a random matrix with i.i.d. Gaussian or Rademacher entries with mean 0 and variance $\sigma^2$. $\mathcal{T} = \{\frac{a-b}{\|a-b\|} : a, b \in \mathcal{X}\}$ be the set of all pair-wise normalized chords. Define metric entropy integral as $\gamma(\mathcal{T}) = \int_0^1 \sqrt{Z(\alpha, \mathcal{T})} d\alpha$, then there exists an absolute constant $c$, such that $\forall \omega, \delta \in (0, 1)$, if $k \geq c\omega^{-2}(\gamma(\mathcal{T})^2 + \log(2/\delta))$, then with probability at least $1 - \delta$, $R$ is $\omega$-isometry on $\mathcal{X}$, namely,*

$$(1 - \omega)k\sigma^2\|x - y\|^2 \leq \|R^T x - R^T y\|^2 \leq (1 + \omega)k\sigma^2\|x - y\|^2, \forall x, y \in \mathcal{X}.$$

Theorem 1 is a generalization of Johnson-Lindenstrauss Lemma, which characterizes the probability of getting a "good" projection matrix with nice isometry property. By a careful analysis under a slightly different assumption on the domain $\mathcal{X}$, we present the generalization bound on compressive NN classifier (learning with $X_R$) in [21] as follows.

**Theorem 2.** *$X \sim \mathcal{X}^n, Y \sim \{0, 1\}^n$ with $X = [x_1, ..., x_n]^T \in \mathbb{R}^{n \times d}$, $x$ is on the unit sphere. Assume that $\eta(x) = Pr(y = 1|x)$ is L-Lipschitz. Let $R \in^{d \times k}$, $k < d$ a random matrix with i.i.d. Gaussian entries following $N(0, 1)$. $(x, y)$ is a test sample with unknown $y$. Denote $(x_R^{(1)}, y_R^{(1)}) \in (X, Y)$ the training sample such that $\frac{1}{\sqrt{k}} R^T x_R^{(1)}$ is the nearest neighbor of $\frac{1}{\sqrt{k}} R^T x$ in the projected space, and the compressive NN classifier $h_R(x) = y_R^{(1)}$. Denote $\mathcal{L}(h^*)$ the risk of Bayes classifier. Then $\forall \omega, \delta \in (0, 1)$, if $k = O(\omega^{-2}(\gamma(\mathcal{T})^2 + \log(2/\delta)))$, with probability $1 - \delta$ we have the risk of compressive NN classifier*

$$E_{X,Y}[\mathcal{L}(h_R(x))] \leq 2\mathcal{L}(h^*(x)) + 2\sqrt{2}(L\sqrt{\frac{1+\omega}{1-\omega}})^{\frac{k}{k+1}}(ne)^{-\frac{1}{k+1}}\sqrt{k}. \tag{2}$$

Equipped with above tools, we are now ready to state our first result on the risk of uniformly quantized compressive nearest neighbor classifier, with finite $n$ and $k$.

**Theorem 3.** *Let $X, Y, R$ and $\eta(x)$ be the same as in Theorem 2. $Q$ is a b-bit uniform quantizer with bin width $\triangle$. Suppose $(x, y)$ is a test sample with unknown $y$. Denote $(x_Q^{(1)}, y_Q^{(1)}) \in (X, Y)$ the training sample such that $\frac{1}{\sqrt{k}} Q(R^T x_Q^{(1)})$ is the nearest neighbor of $\frac{1}{\sqrt{k}} Q(R^T x)$ in the quantized space, and the quantized compressive NN classifier $h_Q(x) = y_Q^{(1)}$. Let $\mathcal{L}(h^*)$ be the risk of Bayes*

*rule. Then $\forall \omega, \delta \in (0,1)$, if $k = O(\omega^{-2}(\gamma(\mathcal{T})^2 + \log(2/\delta)))$, $[-\sqrt{1+\omega}, \sqrt{1+\omega}] \subset [t_0, t_{2^b}]$ and the maximal gap $g_Q \triangleq g_Q(-\sqrt{1+\omega}, \sqrt{1+\omega}) < 2\sqrt{1+\omega}$, then with probability $1-\delta$ over random draws of $R$, the risk of quantized compressive NN classifier is bounded by*

$$E_{X,Y}[\mathcal{L}(h_Q(x))] \le 2\mathcal{L}(h^*(x)) + 2\sqrt{2}(\frac{L\triangle}{g_Q}\sqrt{\frac{1+\omega}{1-\omega}})^{\frac{k}{k+1}}(ne)^{-\frac{1}{k+1}}\sqrt{k} + \frac{2L\triangle\sqrt{k}}{\sqrt{1-\omega}}. \quad (3)$$

**Remark.** *The assumption that $Q$ is uniform quantizer is only for the ease of presentation. For an arbitrary quantizer, the bound also holds with $\triangle$ replaced by a more complicated term.*

The proof involves two interleaving covers of the projected space, which, by Theorem 1, has bounded diameter with high probability. Now we compare Theorem 3 with Theorem 2. Denote the second term in (3) as the random projection error and the last term as quantization error. We observe: 1) The bound preserves sample complexity of $O(e^k)$, which is favorable. 2) The extra quantization error decreases with smaller bin length $\triangle$, which is reasonable since small $\triangle$ implies better approximation to the full-precision RP's in general. 3) When $\triangle \to 0$ which means no quantization applied, we have $g_Q = \triangle$ and the bound reduces to (2) in Theorem 2. Note that, although the factor $\sqrt{k}$ in quantization error term also appears in the RP error term, it implies that the error incurred by quantization becomes larger as $k$ increases. Intuitively, however, large $k$ provides better estimation of the pair-wise angle and thus pair-wise distance (since $\mathcal{X}$ has domain $\mathcal{S}_d$), which should actually reduce the extra loss, since nearest neighbors would be more accurately estimated. This unsatisfactory pattern of quantization error in Theorem 3 comes from the finite sample setting and proof methodology, since the bound is a worst case bound with $n$ and $k$ both finite. Thus, this bound is less meaningful for practical purposes.

**Asymptotic analysis.** Notice that the key difference between NN classifier, compressive NN and quantized compressive NN is simply the space in which we look for the neighbors. More importantly, this procedure essentially depends on the distance estimation. Given that $\mathcal{X}$ is defined on the unit sphere, finding NN in projected or quantized space is identical to finding $x_i \in X$ that has largest estimated cosine between test example $x$. In this case, we do not need to care about the specific space from which we derive the estimator, while the statistical property becomes the major concern.

**Theorem 4.** *(Asymptotic $k$). Let data $X, Y$ and projection matrix $R$ be same as Theorem 3. Let $(x, y)$ be a test sample with unknown $y$. $Q$ is any arbitrary quantizer with increasing reconstruction levels. We estimate the cosine between any two points $s, t \in X$ with $\langle s, t \rangle = \rho_{s,t}$ in the quantized space by $\hat{\rho}_Q(s,t) = \frac{Q(R^T s)^T Q(R^T t)}{k}$. Assume that $\forall s, t \sim \mathcal{X}$, $E[\hat{\rho}_Q(s,t)] = \alpha\rho_{s,t}$ for some $\alpha > 0$. Denote $(x_Q^{(1)}, y_Q^{(1)}) \in (X, Y)$ the training sample such that $\frac{1}{\sqrt{k}}Q(R^T x_Q^{(1)})$ is the nearest neighbor of $\frac{1}{\sqrt{k}}Q(R^T x)$, and the quantized compressive NN classifier $h_Q(x) = y_Q^{(1)}$. Then we have as $k \to \infty$,*

$$E_{X,Y,R}[\mathcal{L}(h_Q(x))] \le E_{X,Y}[\mathcal{L}(h_S(x))] + r_k,$$

*where $r_k = E_{X,x}[\sum_{i:x_i \in \mathcal{G}} \Phi\big(\frac{\sqrt{k}(\cos(x,x_i) - \cos(x,x^{(1)}))}{\sqrt{\xi_{x,x_i}^2 + \xi_{x,x^{(1)}}^2 - 2Corr(\hat{\rho}_Q(x,x_i), \hat{\rho}_Q(x,x^{(1)}))\xi_{x,x_i}\xi_{x,x^{(1)}}}}\big)],$*

*with $\xi_{x,y}^2/k$ the debiased variance of $\hat{\rho}_Q(x,y)$ and $\mathcal{G} = X/x^{(1)}$. $\mathcal{L}(h_S(x))$ is the risk of data space NN classifier, $h_S(x) = y^{(1)}$ with $(x^{(1)}, y^{(1)})$ the nearest neighbor of $x$. $\Phi(\cdot)$ is the CDF of $N(0,1)$.*

**Remark.** *We express the bound in terms of $E_{X,Y}[\mathcal{L}(h_S(x))]$ to highlight the extra quantization error. The assumption that $\hat{\rho}_Q$ has expectation linear in $\rho$ is mainly for the ease of analytical consideration. Similar result also holds in general situations, under additional minor assumptions.*

The bound is intuitive, in the sense that the quantization error term $r_k$ represents the probability of picking different nearest neighbor in data space and quantized space. The benefit of Theorem 4 is that, we factor out $\mathcal{L}(h_S(x))$, instead of $\mathcal{L}(h_R(x))$ as in Theorem 3. Conceptually, we get rid of the error incurred by using the projected space as an intermediate step. The quantization error term $r_k$ is interesting—Note that for $\forall i \in \mathcal{G} = X/x^{(1)}$, $\cos(x,x_i) - \cos(x,x^{(1)}) < 0$ holds. Consequently, when $k \to \infty$, all the $\Phi(\cdot)$ terms in $r_k$ would decrease towards 0 (since $\Phi(t) \to 0$ as $t \to -\infty$). Therefore, we derive a well behaving quantization error term in the asymptotic case: the quantization error indeed decreases with $k$ and converges to that of the data space nearest neighbor classifier.

**Choice of $Q$.** It can be shown that under some mild conditions, small debiased variance ($\xi_{x,x_i}$ and $\xi_{x,x^{(1)}}$) reduces the quantization error $r_k$ in Theorem 3. In addition, by the asymptotic normality of $\hat{\rho}_Q$, given a large $k$ and a query $x$, points near $x^{(1)}$ (i.e., with small $|\cos(x,x_i) - \cos(x, x^{(1)})|$) tend to affect the quantization error more substantially due to the light tail of Gaussian distribution. Hence, for 1-NN classification, we should ideally choose quantizers with low debiased variance around $\rho^* = \cos(x, x_i^{(1)})$, provided that it can be known (or estimated) a priori. In particular, if a quantized estimator has lower debiased variance than the full-precision estimator, then learning with $X_Q$ would outperform learning with $X_R$ in 1-NN classification.

Is there a way to reduce the debiased variance of inner product estimates, for better generalization in NN classification? Recent progress on quantized random projections [28] shows that normalizing the randomly projected vectors (i.e., $R^T x_i, i = 1, ..., n$) can provide smaller debiased variance, especially in high similarity region (large $|\rho|$). This is exactly the situation for most of the NN classifications where $\rho^* = \cos(x, x_i^{(1)})$ is high. More specifically, we can use the estimator

$$\hat{\rho}_{Q,n} = \frac{Q(R^T x_1)^T Q(R^T x_2)}{\|Q(R^T x_1)\| \|Q(R^T x_2)\|}, \tag{4}$$

to estimate $\rho(x_1, x_2)$, instead of the simple inner product estimator $\hat{\rho}_Q = \frac{Q(R^T x_1)^T Q(R^T x_2)}{k}$ used in Theorem 4. We refer interested readers to [28] for more detailed discussions on this topic.

In the following, we derive a corollary regarding the error of compressive NN classifier $h_R(x)$ by noticing that the full-precision RP corresponds to applying quantization with infinite bits.

**Lemma 1.** *Let full-precision linear estimator $\hat{\rho}_R$ be defined as $\hat{\rho}_R(x_1, x_2) = \frac{x_1^T R R^T x_2}{k}, \forall x_1, x_2 \in \mathcal{X}$. Suppose $x, y, z \in \mathbb{R}^d$ are three data points on a unit sphere with inner products $\rho_{xy}, \rho_{xz}$ and $\rho_{yz}$ respectively. Then the covariance*

$$Cov(\hat{\rho}_R(x,y), \hat{\rho}_R(x,z)) = \frac{1}{k}(\rho_{yz} + \rho_{xy}\rho_{xz}).$$

**Corollary 1.** *Let the data $(X, Y)$, $(x, y)$ and projection matrix $R$ be same as Theorem 3, with $Q$ a quantizer with increasing reconstruction levels. We estimate the cosine between any two points $s, t \in X$ with $\langle s, t \rangle = \rho_{s,t}$ in the projected space by $\hat{\rho}_R(s, t) = \frac{s^T R R^T t}{k}$. Denote $(x_R^{(1)}, y_R^{(1)}) \in (X, Y)$ the training sample such that $R^T x_Q^{(1)}$ is the nearest neighbor of $R^T x$ in the projected space, and the NN classifier $h_R(x) = y_R^{(1)}$. Then as $k \to \infty$,*

$$E_{X,Y,R}[\mathcal{L}(h_R(x))] \leq E_{X,Y}[\mathcal{L}(h_S(x))] + r_k,$$

*where $r_k = E_{X,x}[\sum_{i:x_i \in \mathcal{G}} \Phi\big(\frac{\sqrt{k}(\cos(x,x_i) - \cos(x,x^{(1)}))}{\sqrt{(\cos(x,x_i) - \cos(x,x^{(1)}))^2 + 2(1 - \cos(x_i, x^{(1)}))}}\big)]$, with $\mathcal{G} = X/x^{(1)}$.*

## 4 Quantized Compressive Linear Classification with (0,1)-loss

In this section, we consider the generalization error for binary linear classifiers, which include some of the most popular learning models, e.g., logistic regression, linear SVM, etc. Let $\mathcal{H}$ be a hypothesis class of functions on $\mathcal{X} \to \{0, 1\}$. For original data, we assume that a function $H \in \mathcal{H}$ separates $S$ by a hyperplane, and classify each side as a distinct class. Hence, for a test data point $x$, the predicted label returned by $H$ is

$$H(x) = \mathbb{1}\{h^T x > 0\},$$

where $h$ is a vector in $\mathbb{R}^d$ and orthogonal to the separating plane. Since all $x_i$'s are normalized to unit norm, we may assume that $h$ also lies on the unit sphere passing though the origin. The optimal classifier, $\hat{H}$, is the minimizer of (0,1)-loss, defined as

$$\hat{\mathcal{L}}_{(0,1)}(S, h) = \frac{1}{n}\sum_{i=1}^{n} L_{(0,1)}(H(x_i), y_i), \quad L_{(0,1)}(H(x_i), y_i) = \begin{cases} 0, \text{ if } H(x_i) = y_i, \\ 1, \text{ otherwise.} \end{cases} \tag{5}$$

We denote $\hat{h} \in \mathbb{R}^d$ the learned vector associated with $\hat{H}$. $(\hat{H}, \hat{h})$ is called the empirical risk minimization (ERM) classifier. In projected space and quantized space, the ERM classifiers are denoted by similar notation with corresponding subscripts as $\hat{H}_R, \hat{h}_R \in \mathbb{R}^k, \hat{H}_Q$ and $\hat{h}_Q \in \mathbb{R}^k$,

$$\hat{H}(x) = \mathbb{1}\{\hat{h}^T x > 0\}, \quad \hat{H}_R(x) = \mathbb{1}\{\hat{h}_R^T R^T x > 0\}, \quad \hat{H}_Q(x) = \mathbb{1}\{\hat{h}_Q^T Q(R^T x) > 0\}. \tag{6}$$

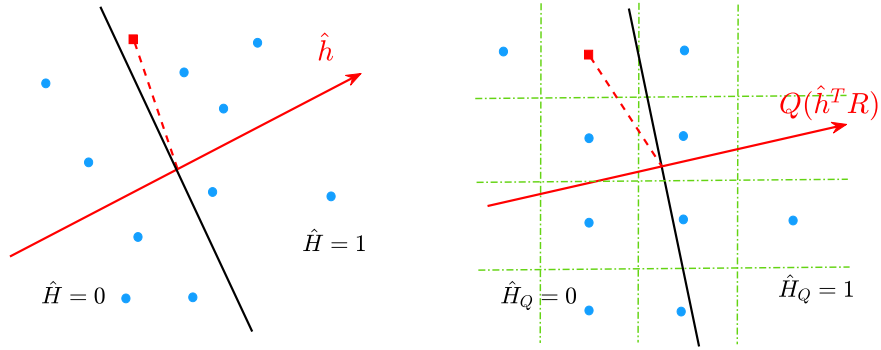

Figure 1: Sign flipping in quantized space. Points on the right of black decision boundary are classified as 1, and 0 otherwise. Green dashed lines are boarders of the quantizer. Left: data space classifier predicts 1. Right: quantized space prediction (using $Q(\hat{h}^T R)$ as predictor) changes to 0.

Now suppose $x$ is a test sample with unknown class $y$, we are interested in the probability of making a wrong prediction by training the classifier in the quantized space,

$$Pr[\hat{H}_Q(x)) \neq y] = E[\mathcal{L}(\hat{H}_Q(x), y)].$$

Existing results on such compressive linear classifier have studied bounds on the same type of objective in the projected space, under finite $k$ setting [13]. Here, we look at this problem in the asymptotic domain. When studying the error incurred by learning in the projected space, an important tool is the following definition.

**Definition 6.** *Let $\hat{h}, x \in \mathbb{R}^d$ be defined above, $\|\hat{h}\| = \|x\| = 1$. Let $\langle \hat{h}, x \rangle = \cos(\hat{h}, x) = \rho > 0$, and $\hat{\rho}_R = \frac{\hat{h}^T R R^T x}{k}$. $R \in \mathbb{R}^{d \times k}$ an i.i.d. standard Gaussian random matrix. The **flipping probability** is defined as*

$$f_k(\rho) = Pr[\rho_R < 0 | \rho > 0]. \tag{7}$$

Intuitively, this quantity measures the probability of changed prediction with the compressed model which learns in the space projected by $R$ when $R^T \hat{h}$ is the classifier. [13] gives the exact formula of this quantity, which reads as

$$f_k(\rho) = \frac{\Gamma(k)}{\Gamma(k/2)^2} \int_0^{\frac{1-\rho}{1+\rho}} \frac{z^{(k-2)/2}}{(1+z)^k} dz = F_{k,k}(\frac{1-\rho}{1+\rho}), \tag{8}$$

where $F$ is the cumulative distribution function (CDF) of F-distribution with $(k, k)$ degrees of freedom. This formula also holds for $\rho < 0$ by simply plugging in $\rho = -\rho$. By symmetry, it suffices to consider $\rho > 0$. As it is well-known that $E[\hat{\rho}_R] = \rho$ and $Var[\hat{\rho}_R] = \frac{1+\rho^2}{k}$, $\hat{\rho}_R$ should asymptotically follow $N(\rho, \frac{1+\rho^2}{k})$ as $k \to \infty$. So the asymptotic flipping probability should be

$$\tilde{f}_k(\rho) = \Phi(-\frac{\sqrt{k}\rho}{\sqrt{1+\rho^2}}). \tag{9}$$

The following proposition confirms this asymptotic convergence.

**Proposition 1.** *As $k \to \infty$, we have $f_k(\rho) \to \tilde{f}_k(\rho)$ for $\rho > 0$.*

For quantized compressive classifier, sign flipping may also happen (an illustrative example is given in Figure 1). By analyzing this event, in the following we state the asymptotic generalization error bound for linear classifiers when working in quantized space instead of data space.

**Theorem 5.** *Let the (0,1)-loss and ERM classifier be defined as (5) and (6). $R \in \mathbb{R}^{d \times k}$ is i.i.d standard normal random matrix. Let $\hat{\mathcal{L}}_{(0,1)}(S, \hat{h}) = \frac{1}{n} \sum_{i=1}^{n} \mathcal{L}_{(0,1)}(H(x_i), y_i)$ be the empirical loss in the data space. $Q$ is a quantizer and the quantized estimator $\hat{\rho}_Q = \frac{Q(R^T s)^T Q(R^T t)}{k}$ has mean $\alpha\rho$,*

$\alpha > 0$, and debiased variance $\xi_\rho^2/k$ at $\rho = \cos(s, t)$, $\forall s, t \sim \mathcal{X}$. Given $(x, y)$ a test sample with $y$ unknown, when $k \to \infty$, with probability at least $1 - 2\delta$ we have

$$Pr[\hat{H}_Q(x) \neq y] \leq \hat{\mathcal{L}}_{(0,1)}(S, \hat{h}) + 2\sqrt{\frac{(k+1)\log\frac{2en}{k+1} + \log\frac{1}{\delta}}{n}}$$

$$+ \frac{1}{n}\sum_{i=1}^{n} f_{k,Q}(\rho_i) + \min\left\{\sqrt{3\log\frac{1}{\delta}}\sqrt{\frac{1}{n}\sum_{i=1}^{n} f_{k,Q}(\rho_i)}, \frac{1-\delta}{\delta n}\sum_{i=1}^{n} f_{k,Q}(\rho_i)\right\},$$

where the flipping probability $f_{k,Q}(\rho_i) = \Phi(-\frac{\sqrt{k}|\rho_i|}{\xi_{\rho_i}})$, with $\rho_i$ the cosine between training sample $x_i$ and ERM classifier $\hat{h}$ in the data space.

In Theorem 5, the first term is the empirical loss in the data space, and the second term is the generic sample complexity in learning theory. The last two terms are called the quantization error. When $b \to \infty$ (full-precision RP's), the bound reduces to that derived in [13] for compressive linear classifier, according to Proposition 1. One important observation is that the quantization error again depends on the debiased variance of the quantized inner product estimator, at different $\rho_i$, $i = 1, ..., n$. This result provides some insights on the influence of quantization for linear classification.

**Choice of $Q$.** Unlike NN classifier, the extra generalization error depends more on the region near 0 for linear classifier. To see this, we notice that the flipping probabilities (8) and (9) decrease as $\rho$ increases. Intuitively, label flipping is much more likely to occur for the points near the boundary (i.e., with small $\hat{h}^T x$). As a result, one may choose a quantizer with small debiased variance around $\rho = 0$ for linear classification. In fact, by the analysis and results from [28], one can show that Lloyd-Max (LM) quantizer gives minimal debiased variance of $\hat{\rho}_Q$ at $\rho = 0$, among all quantizers with same bits. Hence, LM quantization is recommended for linear classification problems.

## 5 Quantized Compressive Least Squares Regression

Compressive least squares (CLS) regression has been studied in several papers, e.g., [30, 20]. [34] shows that in many cases, CLS can match the performance of principle component regression (PCR) but runs faster by avoiding large scale SVD or optimization, especially on high-dimensional data. In CLS, the projected design matrix $X_R$, instead of the original $X$, is used for ordinary least squares (OLS) regression. We are interested in the extra error brought by further quantizing the projections, where $X_Q$ is used as the new design matrix. We call this approach QCLS. In particular, we consider a fix design problem where data $X \in \mathbb{R}^{n \times d}$ is determinant and $Y \in \mathbb{R}^n$ are treated as random. OLS regression with Gaussian error is modeled by

$$Y = X\beta + \epsilon, \tag{10}$$

with $\beta \in \mathbb{R}^d$ and $\epsilon \in \mathbb{R}^n$ contains $i.i.d.$ Gaussian noise with mean 0 and variance $\gamma$. For projected data and quantized data, we also fit a OLS with same response $Y$, while the predictors becomes $\frac{1}{\sqrt{k}}R^T x_i$ and $\frac{1}{\sqrt{k}}Q(R^T x_i)$ respectively. Furthermore, define the expected squared losses as

$$L(\beta) = \frac{1}{n}E_Y[\|Y - X\beta\|^2], \quad L_R(\beta_R) = \frac{1}{n}E_{Y|R}[\|Y - \frac{1}{\sqrt{k}}XR\beta_R\|^2],$$

$$L_Q(\beta_Q) = \frac{1}{n}E_{Y|R}[\|Y - \frac{1}{\sqrt{k}}Q(XR)\beta_Q\|^2]. \tag{11}$$

Note that in the above the expectation is taken $w.r.t.$ $Y$, and $R$ is given. Denote the true minimizers of above losses as $\beta^*$, $\beta_R^*$ and $\beta_Q^*$, respectively. The risk of an estimator in the data space is defined as $r(w) = L(w) - L(\beta^*)$, and analogues $r_R(w_R)$ and $r_Q(w_Q)$ can be also defined in projected and quantized spaces. On the other hand, we have the empirical losses

$$\hat{L}(\beta) = \frac{1}{n}\|Y - X\beta\|^2, \quad \hat{L}_R(\beta_R) = \frac{1}{n}\|Y - \frac{1}{\sqrt{k}}XR\beta_R\|^2, \quad \hat{L}_Q(\beta_Q) = \frac{1}{n}\|Y - \frac{1}{\sqrt{k}}Q(XR)\beta_Q\|^2,$$

$$\tag{12}$$

which are computed from the data. The least squares estimates minimize the empirical losses in a given space, namely,

$$\hat{\beta}^* = \underset{\beta \in \mathbb{R}^d}{\operatorname{argmin}} \hat{L}(\beta), \quad \hat{\beta}_R^* = \underset{\beta \in \mathbb{R}^k}{\operatorname{argmin}} \hat{L}_R(\beta), \quad \hat{\beta}_Q^* = \underset{\beta \in \mathbb{R}^k}{\operatorname{argmin}} \hat{L}_Q(\beta). \tag{13}$$

In particular, $\hat{\beta}$ is the OLS estimator, and $\hat{\beta}_R^*$ and $\hat{\beta}_Q^*$ are called the CLS estimator and QCLS estimator, respectively. The following result bounds the expected loss of $\hat{\beta}_Q^*$, over $Y$ and $R$.

**Theorem 6.** *Let the regression problems be defined as in (10), (11), (12) and (13), with $\gamma$ being the variance of Gaussian noise. Suppose all samples in $X$ has unit norm, $\Sigma = X^T X/n$, and $R \in \mathbb{R}^{d \times k}$ are i.i.d. standard normal with $k < n$. $Q$ is a Lloyd-Max quantizer with distortion $D_Q$ w.r.t. standard Gaussian. Further define $\xi_{2,2} = E[Q(x)^2 x^2]$ with $x \sim N(0,1)$. Then, the expected QCLS risk over loss of data space learner is bounded by*

$$E_{Y,R}[L_Q(\hat{\beta}_Q^*)] - L(\beta^*) \leq \gamma \frac{k}{n} + \frac{1}{k} \|\beta^*\|_\Omega^2, \tag{14}$$

*where $\Omega = [\frac{\xi_{2,2}-1+D_Q}{(1-D_Q)^2} - 1]\Sigma + \frac{1}{1-D_Q} I_d$, with $\|w\|_\Omega = \sqrt{w^T \Omega w}$ the Mahalanobis norm and $I_d$ the identity matrix with rank $d$.*

**Remark.** *Lloyd-Max quantizer is considered for the ease of presentation. Similar result can be derived for general quantizers under extra technical assumptions. When $D_Q = 0$ (no quantization is applied), $\xi_{2,2} = 3$ holds and the bound reduces to the classical bound [20] for CLS.*

**Choice of $Q$.** In Theorem 6, we see that the distortion in general controls the excess risk. In particular, smaller $D_Q$ would reduce expected loss. Although the bound is for LM quantizer, we expect similar results for other quantizers, since smaller distortion in general implies less deviation from the compressed signals. Hence, with a fix number of bits, Lloyd-Max (LM) quantizer, which is built naturally for the purpose of distortion minimization, should be the first choice for QCLS.

## 6 Numerical study

In this section, we validate the theoretical findings through experiments on real-world datasets from UCI repository [12]. Table 1 provides summary statistics, where mean $\rho$ is the average pair-wise cosine of all pairs of samples. Mean 1-NN $\rho$ is the average cosine of each point to its nearest neighbor.

Table 1: Summary statistics of datasets, all standardized to unit norm.

| Dataset | # samples | # features | # classes | Mean $\rho$ | Mean 1-NN $\rho$ |
|---|---|---|---|---|---|
| arcene | 200 | 10000 | 2 | 0.63 | 0.86 |
| BASEHOCK | 1993 | 4862 | 2 | 0.33 | 0.59 |
| orlraws10P | 100 | 10304 | 10 | 0.80 | 0.89 |

**Classification setup.** We test three quantizers: 1-bit Lloyd-Max quantizer, 3-bit Lloyd-Max quantizer and 3-bit uniform quantizer. LM quantizers are optimized w.r.t. standard normal distribution, and the uniform quantizer is symmetric about 0 with $\triangle = 1$, and cut-off points $x = -3.5$ if $x < -3$; $x = 3.5$ if $x > 3$. As discussed in [28], the debiased variance of $\hat{\rho}_Q = \frac{Q(XR)^T Q(XR)}{k}$ cannot be computed exactly. Here we approximate it by simulation as in Figure 2. For 1-NN classification, we take each data point as test sample and the rest as training data over all the examples, and report the mean test accuracy. For linear classifier, we feed the inner product estimation matrix $X_Q X_Q^T$ as the kernel matrix into a linear SVM

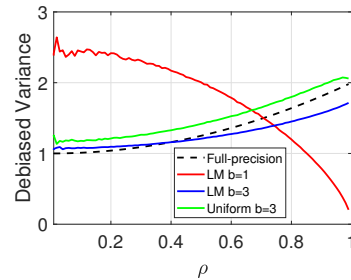

Figure 2: Empirical debiased variance. To be divided by $k$.

solver [5]. We randomly split the data to 60% for training and 40% for testing, and the best test accuracy among all hyper-parameter $C$ is reported, averaged over 5 repetition's.

**Linear SVM.** At $\rho = 0$, Figure 2 shows that the debiased variances of estimators using different quantizers admit the order 1-bit LM>3-bit uniform>3-bit LM>full-precision. Therefore, following the discussion in Theorem 5, we expect test error in the same order, which is confirmed by Figure 5.

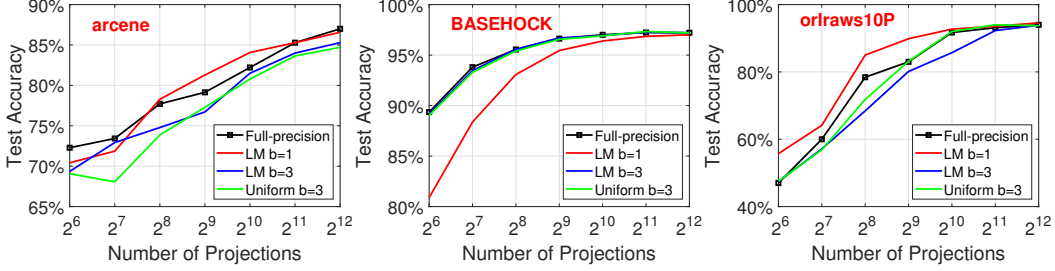

Figure 4: Test accuracy of quantized compressive nearest neighbor classification.

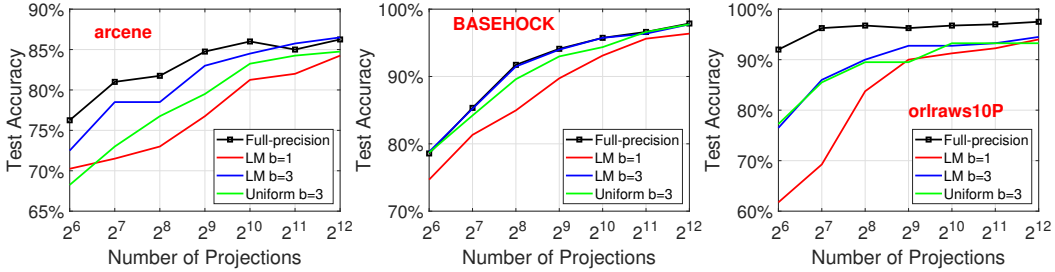

Figure 5: Test accuracy of quantized compressive linear SVM.

**NN classification.** Theorem 4 states that small debiased variance around the "mean 1-NN $\rho$" should be beneficial for 1-NN classification. *BASEHOCK* dataset has mean 1-NN $\rho = 0.59$, the point at which the debiased variance is compared as 1-bit LM $>$ 3-bit uniform $>$ full-precision $\approx$ 3-bit LM. Hence, we see in Figure 4 that the NN classification error is in the same sequence on this dataset. On the other hand, the mean 1-NN $\rho$ of *arcene* and *orlraws10P* is high (around 0.9). At this point, 1-bit LM quantizer has much smaller debiased variance than others. Therefore, we expect 1-bit LM to provide highest test accuracy on these two datasets, which is again consistent with Figure 4. In conclusion,

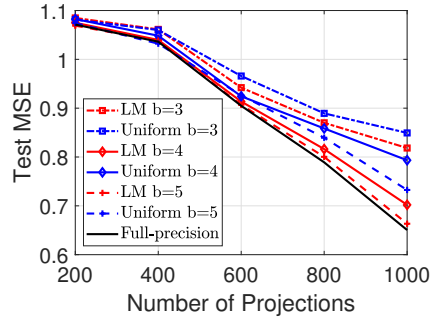

Figure 3: Test MSE of QCLS.

our empirical observations validate the theoretical results and analysis in Theorem 4 and Theorem 5 on the influence of debiased estimator variance on NN and linear classifiers, at different $\rho$ level.

**Simulated QCLS.** We simulate data $X \in \mathbb{R}^{3000 \times 1200}$ and $\beta$ both following *i.i.d.* $N(0,1)$, and noise $\epsilon \sim N(0, 0.2)$. We compare LM quantizers with equal-bit uniform quantizers, for $b = 3, 4, 5$. The distortion is $(0.035, 0.009, 0.002)$ for LM quantizers and $(0.043, 0.026, 0.019)$ for uniform quantizers. In Figure 3, we see that the order of test MSE perfectly agrees with the order of distortion from high to low, and LM quantizers always outperform uniform quantizers with same bits. As the distortion gets smaller, the performance of QCLS approaches that of CLS. These observations verify the conclusion in Theorem 6 that quantizers with smaller distortion generalize better for QCLS.

# 7   Concluding Remarks

This paper studies the generalization error of various quantized compressive learning models, including nearest neighbor classifier, linear classifier and linear regression. Our theoretical results provide useful guidance for choosing appropriate quantizers for different models, which in particular depicts an interesting connection between debiased variance of inner product estimates and the generalization performance on classification tasks. Quantizers with small debiased variance are favorable for NN classifier and linear classifier, in high similarity region and around $\rho = 0$, respectively. For linear regression, quantizers with smaller distortion tend to perform better. As a consequence, Lloyd-Max (LM) quantizer is recommended for both linear classification and regression, and normalizing the projections may help with nearest neighbor classification. Our work contributes to understanding the underlying statistical aspects of learning with quantized random projections, and provides useful implications to various machine learning applications where data compression is useful.

## Footnotes

[1]The work of Xiaoyun Li was conducted during the internship at Baidu Research.

[2]Instance normalization is a standard data preprocessing step for many learning models. In this paper, this assumption is mainly for convenience. Our analysis can be modified for scenarios without data normalization.

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
