[Supplementary Material · RP-Error-Supp.pdf]

# Supplemental Materials for: Generalization Error Analysis of Quantized Compressive Learning

**Xiaoyun Li**[*]
Department of Statistics
Rutgers University
Piscataway, NJ 08854, USA
xiaoyun.li@rutgers.edu

**Ping Li**
Cognitive Computing Lab
Baidu Research
Bellevue, WA 98004, USA
liping11@baidu.com

## A  Proofs of Theorems

### A.1  Technical Lemmas

**Lemma A1.** *[2] Let the losses and estimators be defined as in Theorem 6 under fixed design setting. Let $\gamma = Var[y_i]$, $i = 1, ..., n$, and $X$ is fixed. Then the expected risk*

$$E_Y[L(\hat{\beta}^*)] - L(\beta^*) \leq \gamma \frac{rank(X)}{n}.$$

**Lemma A2.** *[6] Let $X = \sum_{i=1}^n X_i$ with possibly dependent $X_i$'s. $Y = \sum_{i=1}^n Y_i$ where $Y_i$'s are independent copies of $X_i$'s (i.e. $Y_i$ has same distribution as $X_i$, $i = 1, ..., n$). If $B$ is a Chernoff bound on $Pr[Y - E[Y] \geq \epsilon]$, then we have*

$$Pr[X - E[X] \geq \epsilon] \leq B^{\frac{1}{n}}.$$

**Lemma A3.** *[4] Let $(x, y)$ follows standard bi-variate normal distribution with unit variance and covariance $\rho$. $Q$ is a Lloyd-Max quantizer with distortion $D_Q$. Then,*

$$E[xQ(y)] = (1 - D_Q)\rho.$$

### A.2  Proof of Theorem 3

*Proof.* The proof idea is similar to [3], but we operate in the quantized space which is more complicated. First, we have

$$E_{X,Y}[\mathcal{L}(h_Q(x))] = E_{X \sim \mathcal{X}, Y \sim \eta(X)}[E_{x \sim \mathcal{X}, y \sim \eta(x)}[\mathbb{1}\{h_Q(x) \neq y\}|X, Y]]$$

$$= E_{X \sim \mathcal{X}, x \sim \mathcal{X}}[Pr_{y \sim \eta(x), y_Q^{(1)} \sim \eta(x_Q^{(1)})}[y_Q^{(1)} \neq y|X, x]]. \tag{1}$$

We can bound the inner probability for any two points $x, x' \sim \mathcal{X}$ as

$$Pr_{y \sim \eta(x), y' \sim \eta(x')}[y \neq y'|x, x'] = \eta(x)(1 - \eta(x')) + \eta(x')(1 - \eta(x))$$

$$= 2\eta(x)(1 - \eta(x)) + (\eta(x) - \eta(x'))(2\eta(x) - 1)$$

$$\leq 2\eta(x)(1 - \eta(x)) + \|\eta(x) - \eta(x')\|, \tag{2}$$

by the definition of $\eta(x)$. Here we use the fact $|2\eta(x) - 1| \leq 1$. Combining (2) and (1) we have

$$E_{X,Y}[\mathcal{L}(h_Q(x))] \leq E_{X \sim \mathcal{X}, x \sim \mathcal{X}}[2\eta(x)(1 - \eta(x)) + \|\eta(x) - \eta(x_Q^{(1)})\|].$$

Notice from a classical result that

$$L(h^*(x)) = \min\{\eta(x), 1 - \eta(x)\},$$

---

[*]The work of Xiaoyun Li was conducted during the internship at Baidu Research.

we obtain

$$E_{X,Y}[\mathcal{L}(h_Q(x))] \leq 2\mathcal{L}(h^*(x)) + E_{X,Y,x\sim\mathcal{X}}[\|\eta(x) - \eta(x_Q^{(1)})\|].$$

The first term is the Bayes risk, and it remains to bound the second term. By Theorem 1, given that $k = O(\omega^{-2}(\gamma(\mathcal{T})^2 + \log(2/\delta)))$, with probability $1 - \delta$ we have

$$(1 - \omega)\|x - y\|^2 \leq \|\frac{1}{\sqrt{k}}R^T x - \frac{1}{\sqrt{k}}R^T y\|^2 \leq (1 + \omega)\|x - y\|^2, \forall x, y \in \mathcal{X}. \tag{3}$$

Denote this event $\Omega$. Now we proceed our analysis in $\Omega$ (with high probability). The space of projected samples $S_R = \{\frac{1}{\sqrt{k}}R^T x_1, ..., \frac{1}{\sqrt{k}}R^T x_n\}$ is now bounded by

$$u_R = \|\frac{1}{\sqrt{k}}R^T x\| \leq \sqrt{1 + \omega},$$

by taking $y = 0$ in (3). Therefore, $S_R \subset [-u_R, u_R]^k$. Now we cover $[-u_R, u_R]$ by $N_C = (2u_R/\epsilon)^k$ boxes with length $\epsilon$. For the test sample $x$, let $B_\epsilon(x)$ be the box containing $\frac{1}{\sqrt{k}}R^T x$. In the event $\Omega$, we have

$$E_{X,Y,x}[\|\eta(x) - \eta(x_Q^{(1)})\| | \Omega] = E_{X,Y,x}[\|\eta(x) - \eta(x_Q^{(1)})\| | \Omega, V]Pr(V)$$
$$+ E_{X,Y,x}[\|\eta(x) - \eta(x_Q^{(1)})\| | \Omega, V^c]Pr(V^c),$$

where the event $V = \{B_\epsilon(x) \cap S_R(X) = \emptyset\}$, $V^c$ its complement, with $S_R(X) = \{\frac{1}{\sqrt{k}}R^T x_1, ..., \frac{1}{\sqrt{k}}R^T x_n\}$ the projected samples. By Lemma 19.2 in [5], we have

$$Pr(V) \leq N_C/ne.$$

Thus,

$$E_{X,Y,x}[\|\eta(x) - \eta(x_Q^{(1)})\| | \Omega] \leq \frac{N_C}{ne} + E_{X,Y,x}[\|\eta(x) - \eta(x_Q^{(1)})\| | \Omega, V^c]$$
$$\leq \frac{N_C}{ne} + L \cdot E_{X,Y,x}[\|x - x_Q^{(1)}\| | \Omega, V^c]$$
$$\leq \frac{N_C}{ne} + \frac{L}{\sqrt{1 - \omega}}E_{X,Y,x}[\|\frac{1}{\sqrt{k}}R^T x - \frac{1}{\sqrt{k}}R^T x_Q^{(1)}\| | V^c],$$

where the second line is because $\eta(x)$ is $L$-Lipschitz and the last line is due to (3).

Figure 1: An illustration of bounding the distance in projected space $S_R$ with two covers. For simplicity we omit the scaling term $\frac{1}{\sqrt{k}}$. Boxes resulted from red dash lines are the $\epsilon$- cover constructed by hand, and boxes surrounded by blue solid lines are induced cover by $Q$.

To bound the second term, we consider another cover of $S_R$ which is intrinsically induced by the borders of $Q$. Denote $x_R^{(1)}$ as the nearest point of $x$ in the projected space. In this case, we know

that $\|\frac{1}{\sqrt{k}}R^T x - \frac{1}{\sqrt{k}}R^T x_R^{(1)}\| \le \epsilon\sqrt{k}$. However, $\|\frac{1}{\sqrt{k}}R^T x - \frac{1}{\sqrt{k}}R^T x_Q^{(1)}\|$ cannot be bounded in this way, due to the discretization of quantizing function $Q$. In a simple example (Figure 1), assume we only have 3 points in a 2-D case. Denote the centroids of these 3 points respectively as $\mu(\frac{1}{\sqrt{k}}R^T x)$, $\mu(\frac{1}{\sqrt{k}}R^T x_R^{(1)})$ and $\mu(\frac{1}{\sqrt{k}}R^T x_Q^{(1)})$. In this plot, $V^c$ is obviously satisfied since there are two points in the same $\epsilon$-box. Now the green point $(\frac{1}{\sqrt{k}}R^T x_R^{(1)})$ is closer to black point $(\frac{1}{\sqrt{k}}R^T x_R^{(1)})$ in the space of $S_R$, but after quantization, the nearest neighbor returned changes to the pink point $(\frac{1}{\sqrt{k}}R^T x_Q^{(1)})$, since $\mu(\frac{1}{\sqrt{k}}R^T x_Q^{(1)})$ lies closer to $\mu(\frac{1}{\sqrt{k}}R^T x)$ than $\mu(\frac{1}{\sqrt{k}}R^T x_R^{(1)})$. However, $\|\frac{1}{\sqrt{k}}R^T x - \frac{1}{\sqrt{k}}R^T x_Q^{(1)}\|$ might be greater than $\epsilon\sqrt{2}$.

Note that for uniform quantizer, the distances between nearby reconstruction levels all equal to $\triangle$, and the distances between consecutive borders (view $-\sqrt{1-\omega}$ and $\sqrt{1-\omega}$ as borders too) are upper bounded by $g_Q(-\sqrt{1-\omega},\sqrt{1-\omega})$. Using triangle inequality, we get

$$[\|\frac{1}{\sqrt{k}}R^T x - \frac{1}{\sqrt{k}}R^T x_Q^{(1)}\||V^c] \le [\|\frac{1}{\sqrt{k}}R^T x - \mu(\frac{1}{\sqrt{k}}R^T x)\| + \|\frac{1}{\sqrt{k}}R^T x_Q^{(1)} - \mu(\frac{1}{\sqrt{k}}R^T x_Q^{(1)})\|$$

$$+ \|\mu(\frac{1}{\sqrt{k}}R^T x) - \mu(\frac{1}{\sqrt{k}}R^T x_Q^{(1)})\||V^c]$$

$$\le \frac{\triangle\sqrt{k}}{2} + \frac{\triangle\sqrt{k}}{2} + \|\mu(\frac{1}{\sqrt{k}}R^T x) - \mu(\frac{1}{\sqrt{k}}R^T x_R^{(1)})\||V^c].$$

To bound the last term, we need to find out that given two points $a, b$ in a same $\epsilon$-box, how large the distance between their quantized centroids $\mu(a)$, $\mu(b)$ can be. We proceed by noticing that for a given $B_\epsilon(x)$ with any $\epsilon$, the maximum number of different $Q$-boxes that can be contained (perhaps partially) on the diagonal of $B_\epsilon(x)$ is equal to $\lfloor\frac{\epsilon}{g_Q}\rfloor + 2$. The largest distance between the centroids occurs when two points fall into the two regions on the diagonal endpoints (as black and green stars in Figure 1), which equals to $(\lfloor\frac{\epsilon}{g_Q}\rfloor + 1)\triangle\sqrt{k}$. Therefore, we have

$$\|\mu(\frac{1}{\sqrt{k}}R^T x) - \mu(\frac{1}{\sqrt{k}}R^T x_R^{(1)})\||V^c] \le (\lfloor\frac{\epsilon}{g_Q}\rfloor + 1)\triangle\sqrt{k} \le (\frac{\epsilon\triangle}{g_Q} + \triangle)\sqrt{k},$$

where for simplicity we write $g_Q$ instead of $g_Q(-\sqrt{1-\omega},\sqrt{1-\omega})$. Hence, we get the worst case bound

$$E_{X,Y,x}[\|\frac{1}{\sqrt{k}}R^T x - \frac{1}{\sqrt{k}}R^T x_Q^{(1)}\||V^c] \le (\frac{\epsilon\triangle}{g_Q} + 2\triangle)\sqrt{k}.$$

Combining with previous result, we obtain

$$E_{X,Y,x}[\|\eta(x) - \eta(x_Q^{(1)})\||\Omega] \le \frac{N_C}{ne} + \frac{L\triangle\epsilon\sqrt{k}}{g_Q\sqrt{1-\omega}} + \frac{2L\triangle\sqrt{k}}{\sqrt{1-\omega}}.$$

Now we choose $\epsilon$ to minimize the RHS. Let $f(\epsilon) = \frac{(\sqrt{1+\omega}/\epsilon)^k}{ne} + \frac{L\triangle\epsilon\sqrt{k}}{g_Q\sqrt{1-\omega}}$. Following the standard technique, we take the derivative for $f$ with respect to $\epsilon$ and set it to zero, which yields

$$\epsilon^* = (2\sqrt{1+\omega})^{\frac{k}{k+1}}(\frac{L\triangle}{g_Q\sqrt{1-\omega}}ne)^{-\frac{1}{k+1}}\sqrt{k}^{\frac{1}{k+1}}.$$

Plugging in the expression and after some calculation we get

$$E_{X,Y,x}[\|\eta(x) - \eta(x_Q^{(1)})\||\Omega] \le (\frac{2L\triangle}{g_Q}\sqrt{\frac{1+\omega}{1-\omega}})^{\frac{k}{k+1}}(ne)^{-\frac{1}{k+1}}\sqrt{k}(\sqrt{k}^{-\frac{2k+1}{k+1}} + \sqrt{k}^{\frac{1}{k+1}}).$$

Following [3], we have $2^{\frac{k}{k+1}}(\sqrt{k}^{-\frac{2k+1}{k+1}} + \sqrt{k}^{\frac{1}{k+1}}) \le 2\sqrt{2}$. Replacing the terms and combining all parts together, the proof is complete.

$$\square$$

## A.3 Proof of Theorem 4

*Proof.* The proof is based on the probability that $x_Q^{(1)}$ is different from $x^{(1)}$. First we have

$$
\begin{aligned}
E_{X,Y,R}[\mathcal{L}(h_Q(x))] &= E_{X,Y}[E_{x,y,R}[\mathbb{1}\{h_Q(x) \neq y\}|X,Y]] \\
&= E_{X,Y}\{E_{x,y,R}[\mathbb{1}\{h_S(x) \neq y\}\mathbb{1}\{h_Q(x) = h_S(x)\} \\
&\qquad\qquad + \mathbb{1}\{h_S(x) = y\}\mathbb{1}\{h_Q(x) \neq h_S(x)\}|X,Y]\} \\
&\leq E_{X,Y}\{E_{x,y,R}[\mathbb{1}\{h_S(x) \neq y\} + \mathbb{1}\{h_Q(x) \neq h_S(x)\}|X,Y]\} \\
&\triangleq A + B.
\end{aligned}
$$

We recognize the term $A$ is simply the risk of data space NN classifier, $A = E_{X,Y}[\mathcal{L}(h_S(x))]$. It suffices to study term $B$. Note that

$$
\begin{aligned}
B &= E_{X \sim \mathcal{X}, x \sim \mathcal{X}, y^{(1)} \sim \eta(x^{(1)}), y_Q^{(1)} \sim \eta(x_Q^{(1)}), R}\mathbb{1}\{y_Q^{(1)} \neq y^{(1)}\}] \\
&= E_{X,x}\{Pr_{y^{(1)} \sim \eta(x^{(1)}), y_Q^{(1)} \sim \eta(x_Q^{(1)}), R}[x_Q^{(1)} \neq x^{(1)}, y_Q^{(1)} \neq y^{(1)}|X,x]\} \\
&\leq E_{X,x}\{Pr_R[x_Q^{(1)} \neq x^{(1)}|X,x]\} \\
&\triangleq E_{X,x}\{P_c\},
\end{aligned}
$$

where the second line is because $y_R^{(1)} \neq y^{(1)}$ implies that $x_R^{(1)} \neq x^{(1)}$. For a fixed $X$ and $x$, we denote the set $\mathcal{G} = X/x^{(1)}$. Then for the inner probability, we have

$$
\begin{aligned}
P_c &= \sum_{i:x_i \in \mathcal{G}} Pr_R[x_Q^{(1)} = x_i|X,x] \\
&= \sum_{i:x_i \in \mathcal{G}} Pr[\bigcap_{x_j \neq x_i} \{\hat{\rho}_Q(x,x_i) \geq \hat{\rho}_Q(x,x_j)\}|X,x] \\
&\leq \sum_{i:x_i \in \mathcal{G}} Pr[\hat{\rho}_Q(x,x_i) \geq \hat{\rho}_Q(x,x^{(1)})|X,x] \qquad (4)
\end{aligned}
$$

due to the equivalence of inner product and Euclidean distance estimation. Under the asymptotic assumption $k \to \infty$, by Central Limit Theorem (CLT) we know that for any $x, y \in X$,

$$
\hat{\rho}_Q(x,y) \sim N(\alpha\rho_{x,y}, \frac{\sigma_{x,y}^2}{k}),
$$

for $\sigma_{x,y} = Var[Q(r^Tx)^TQ(r^Ty)]$ a fixed constant given $x,y$. Here $r$ is a column of $R$. Next, we obtain for $\forall i,j$,

$$
\hat{\rho}_Q(x,x_i) - \hat{\rho}_Q(x,x_j) \sim N(\alpha(\rho_{x,x_i} - \rho_{x,x_j}), \sigma_{x,x_i}^2 + \sigma_{x,x_j}^2 - 2Corr(\hat{\rho}_Q(x,x_i)\hat{\rho}_Q(x,x_j)\sigma_{x,x_i}\sigma_{x,x_j}).
$$

Therefore,

$$
\begin{aligned}
Pr[\hat{\rho}_Q(x,x_i) \geq \hat{\rho}_Q(x,x_j)] &= Pr[\hat{\rho}_Q(x,x_i) - \hat{\rho}_Q(x,x_j) \geq 0] \\
&= \Phi\Big(\frac{\sqrt{k}\alpha(\rho_{x,x_i} - \rho_{x,x_j})}{\sqrt{\sigma_{x,x_i}^2 + \sigma_{x,x_j}^2 - 2Corr(\hat{\rho}_Q(x,x_i), \hat{\rho}_Q(x,x_j))\sigma_{x,x_i}\sigma_{x,x_j}}}\Big) \\
&= \Phi\Big(\frac{\sqrt{k}(\rho_{x,x_i} - \rho_{x,x_j})}{\sqrt{\xi_{x,x_i}^2 + \xi_{x,x_j}^2 - 2Corr(\hat{\rho}_Q(x,x_i), \hat{\rho}_Q(x,x_j))\xi_{x,x_i}\xi_{x,x_j}}}\Big),
\end{aligned}
$$

since by the definition of debiased variance we have $\xi_{x,x_i}^2 = \frac{\sigma_{x,x_i}^2}{\alpha^2}$. Now plugging above equation into (4), we have

$$
B = E_{X,x}[\sum_{i:x_i \in \mathcal{G}} \Phi\Big(\frac{\sqrt{k}(\cos(x,x_i) - \cos(x,x^{(1)}))}{\sqrt{\xi_{x,x_i}^2 + \xi_{x,x^{(1)}}^2 - 2Corr(\hat{\rho}_Q(x,x_i), \hat{\rho}_Q(x,x^{(1)}))\xi_{x,x_i}\xi_{x,x^{(1)}}}}\Big)],
$$

by noting that $\rho_{x,x_i} = \cos(x,x_i)$. Combining parts together, we get the result as required.

$\square$

## A.4 Proof of Lemma 1

*Proof.* Denote the random projection matrix $R \in \mathbb{R}^{d \times k}$. Recall that the estimates of $\rho_{xy}$ and $\rho_{xz}$ are

$$\hat{\rho}_R(x, y) = \frac{x^T RR^T y}{k}, \quad \hat{\rho}_R(x, z) = \frac{x^T RR^T z}{k}.$$

Denote the columns of $R$ as $[r_1, ..., r_k]$, we have

$$E[\hat{\rho}_R(x, y)\hat{\rho}_R(x, z)]$$

$$= \frac{1}{k^2} E[xRR^T y^T xRR^R z^T]$$

$$= \frac{1}{k^2} [\langle x, r_1 \rangle, ..., \langle x, r_k \rangle] \begin{bmatrix} \langle y, r_1 \rangle \\ \vdots \\ \langle y, r_k \rangle \end{bmatrix} [\langle x, r_1 \rangle, ..., \langle x, r_k \rangle] \begin{bmatrix} \langle z, r_1 \rangle \\ \vdots \\ \langle z, r_k \rangle \end{bmatrix}$$

$$= \frac{1}{k^2} (\sum_{i=1}^{k} \langle x, r_1 \rangle \langle y, r_1 \rangle) \cdot (\sum_{i=1}^{k} \langle x, r_1 \rangle \langle z, r_1 \rangle)$$

$$= \frac{1}{k^2} [\sum_{i=1}^{k} (\sum_{p=1}^{d} x_p r_{ip})(\sum_{q=1}^{d} y_q r_{iq})] \cdot [\sum_{j=1}^{k} (\sum_{s=1}^{d} x_s r_{js})(\sum_{t=1}^{d} y_t r_{jt})]$$

$$= \frac{1}{k^2} \sum_{i=1}^{k} \sum_{j=1}^{k} [\sum_{p=1}^{d} \sum_{q=1}^{d} \sum_{s=1}^{d} \sum_{t=1}^{d} x_p y_q x_s z_t E[r_{ip} r_{iq} r_{js} r_{jt}]]$$

$$= \frac{1}{k^2} \{\sum_{i=1}^{k} \sum_{j \neq i}^{k} [\sum_{p=1}^{d} \sum_{q=1}^{d} \sum_{s=1}^{d} \sum_{t=1}^{d} x_p y_q x_s z_t E[r_{ip} r_{iq} r_{js} r_{jt}]] + \sum_{i=1}^{k} [\sum_{p=1}^{d} \sum_{q=1}^{d} \sum_{s=1}^{d} \sum_{t=1}^{d} x_p y_q x_s z_t E[r_{ip} r_{iq} r_{is} r_{it}]]\}$$

$$\triangleq A + B.$$

For the first term $A$, since $i \neq j$ and all entries of $R$ are *i.i.d.* standard normal, the expectation is non-zero only when $p = q$ and $s = t$. Also note the each row vector $r_i$ and $r_j$ are independent. Consequently we obtain

$$A = \frac{k(k-1)}{k^2} (\sum_{p=1}^{d} \sum_{s=1}^{d} x_p y_p x_s z_s) = \frac{k-1}{k} \langle x, y \rangle \cdot \langle x, z \rangle = \frac{k-1}{k} \rho_{xy} \rho_{xz}.$$

For term $B$, we note that the expectation is non-zero when: (i) $p = q$ and $s = t$; (ii) $p = s$ and $q = t$; or (iii) $p = t$ and $q = s$. In these cases, when $p, q, s, t$ are not all equal, the expected value is simply $E[r_{ip}^2 r_{iq}^2] = 1$. When $p = q = s = t$, the expected value is $E[r_{ip}^4] = 3$. Therefore we have

$$B = \frac{k}{k^2} \{\sum_{p=1}^{d} \sum_{s=1}^{d} x_p y_p x_s z_s + \sum_{p=1}^{d} \sum_{q=1}^{d} x_p y_q x_p z_q + \sum_{p=1}^{d} \sum_{q=1}^{d} x_p y_q x_q z_p - 2 \times 3 \sum_{p=1}^{d} x_p y_p x_p z_p\}$$

$$= \frac{1}{k} [\langle x, y \rangle \cdot \langle x, z \rangle + \|x\|^2 \langle y, z \rangle + \langle x, y \rangle \cdot \langle x, z \rangle + 3 \times 2 \sum_{p=1}^{d} x_p y_p x_p z_p - 2 \times 3 \sum_{p=1}^{d} x_p y_p x_p z_p]$$

$$= \frac{1}{k} (\rho_{yz} + 2\rho_{xy} \rho_{xz}),$$

where the first line is due to the fact that we count the case $p = q = s = t$ for three times. Now putting parts together, we have

$$Cov(\hat{\rho}_R(x, y), \hat{\rho}_R(x, z)) = \frac{1}{k^2} E[x^T RR^T y x^T RR^R z] - E[\hat{\rho}_R(x, y)]E[\hat{\rho}_R(x, z)]$$

$$= \frac{(k-1)\rho_{xy} \rho_{xz} + \rho_{yz} + 2\rho_{xy} \rho_{xz}}{k} - \rho_{xy} \rho_{xz}$$

$$= \frac{1}{k} (\rho_{yz} + \rho_{xy} \rho_{xz}).$$

$\square$

## A.5 Proof of Proposition 1

*Proof.* To start with, we notice that for $x_i$, $y_i$, $i = 1, ..., k$ all *i.i.d.* standard normal,

$$F_{k,k}\left(\frac{1-\rho}{1+\rho}\right) = Pr\left[\sum_{i=1}^{k} x_i^2 \le \frac{1-\rho}{1+\rho}\sum_{i=1}^{k} y_i^2\right] = Pr\left[\frac{1}{k}\sum_{i=1}^{k} x_i^2 \le \frac{1-\rho}{1+\rho}\left(\frac{1}{k}\sum_{i=1}^{k} y_i^2\right)\right].$$

By Central Limit Theorem we have $w = \frac{1}{k}\sum_{i=1}^{k} x_i^2 \sim N(1, 2/k)$, $z = \frac{1}{k}\sum_{i=1}^{k} y_i^2 \sim N(1, 2/k)$ and they are independent. Hence, when $k \to \infty$, we have

$$\begin{aligned}
f_k(\rho) = & F_{k,k}\left(\frac{1-\rho}{1+\rho}\right) = Pr\left[w \le \frac{1-\rho}{1+\rho}z\right]\\
= & \int_{-\infty}^{\infty} \frac{\sqrt{k}}{2\sqrt{\pi}}e^{-\frac{k(z-1)^2}{4}}\int_{-\infty}^{\frac{1-\rho}{1+\rho}z} \frac{\sqrt{k}}{2\sqrt{\pi}}e^{-\frac{k(w-1)^2}{4}}dwdz\\
= & \int_{-\infty}^{\infty} \frac{\sqrt{k}}{2\sqrt{\pi}}e^{-\frac{k(z-1)^2}{4}}\Phi\left(\sqrt{\frac{k}{2}}\left(\frac{1-\rho}{1+\rho}z-1\right)\right)dz\\
= & \int_{-\infty}^{\infty} \frac{1}{\sqrt{2\pi}}e^{-\frac{s^2}{2}}\Phi\left(\frac{(1-\rho)s-\sqrt{2k}\rho}{1+\rho}\right)ds\\
= & E_s\left[\Phi\left(\frac{(1-\rho)s-\sqrt{2k}\rho}{1+\rho}\right)\right],
\end{aligned}$$

where the second and third line are derived by simple change of variable, and $s \sim N(0,1)$. For another $v \sim N(0,1)$ independent of $s$, by law of total expectation we obtain

$$\begin{aligned}
& E_s\left[\Phi\left(\frac{(1-\rho)s-\sqrt{2k}\rho}{1+\rho}\right)\right]\\
= & E_{s,v}\left[\mathbb{1}\left\{v \le \frac{(1-\rho)s-\sqrt{2k}\rho}{1+\rho}\right\}\right]\\
= & Pr\left[v - \frac{1-\rho}{1+\rho}s \le -\frac{\sqrt{2k}\rho}{1+\rho}\right]\\
= & Pr\left[\frac{v - \frac{(1-\rho)}{1+\rho}s}{\sqrt{1+\frac{(1-\rho)^2}{(1+\rho)^2}}} \le -\frac{\sqrt{2k}\rho}{(1+\rho)\sqrt{1+\frac{(1-\rho)^2}{(1+\rho)^2}}}\right]\\
= & \Phi\left(-\frac{\sqrt{k}\rho}{\sqrt{1+\rho^2}}\right) = \tilde{f}_k(\rho).
\end{aligned}$$

This completes the proof. $\square$

## A.6 Proof of Theorem 5

*Proof.* The proof follows from [1]. First by classical VC theory [7], with probability $1 - \delta$ we have

$$Pr[\hat{H}_Q(x)) \ne y] \le \hat{\mathcal{L}}_{(0,1)}(S_Q, \hat{h}_Q) + 2\sqrt{\frac{(k+1)\log\frac{2en}{k+1} + \log\frac{1}{\delta}}{n}},$$

where $\hat{\mathcal{L}}_{(0,1)}(S_Q, \hat{h}_Q) = \frac{1}{n}\sum_{i=1}^{n} \mathcal{L}_{(0,1)}(\hat{H}_Q(Q(R^T x_i)), y_i)$ the empirical loss in the quantized space (with optimal ERM quantizer $\hat{h}_Q$ in $S_Q$). Since $\hat{h}_Q$ is the minimizer of $\hat{\mathcal{L}}_{(0,1)}(S_Q, \hat{h}_Q)$, we

have

$$\hat{\mathcal{L}}_{(0,1)}(S_Q, \hat{h}_Q) \le \hat{\mathcal{L}}_{(0,1)}(S_Q, Q(R^T\hat{h}))$$
$$= \hat{\mathcal{L}}_{(0,1)}(S, \hat{h}) + (\hat{\mathcal{L}}_{(0,1)}(S_Q, Q(R^T\hat{h})) - \hat{\mathcal{L}}_{(0,1)}(S, \hat{h}))$$
$$\le \hat{\mathcal{L}}_{(0,1)}(S, \hat{h}) + \frac{1}{n}\sum_{i=1}^{n} \mathbb{1}\{sign(Q(\hat{h}^T R)Q(R^T x_i)) \ne sign(\hat{h}^T x_i)\}$$
$$:= \le \hat{\mathcal{L}}_{(0,1)}(S, \hat{h}) + M.$$

We note that $M$ is a sum of dependent flipping probabilities because of the commonly used projection matrix $R$. Using Markov's Inequality we have

$$M \le (1 + \frac{1 - \delta}{\delta})E_R[M]$$

with probability $1 - \delta$. To get a better bound with small $\delta$, we make use of Lemma A2. By applying the lemma, if $M^*$ is an independent copy of $M$, standard Chernoff bound gives

$$Pr[M^* \ge (1 + \epsilon)E_R[M^*]] \le \exp(-nE_R[M^*]\epsilon^2/3).$$

Then, Lemma A2 yields

$$Pr[M \ge (1 + \epsilon)E_R[M]] \le \exp(-nE_R[M^*]\epsilon^2/3)^{\frac{1}{n}}$$
$$= \exp(-E_R[M]\epsilon^2/3).$$

Transforming probability bound to expectation bound, we obtain with probability $1 - \delta$,

$$M \le E_R[M] + \sqrt{3E_R[M]\log\frac{1}{\delta}}.$$

The proof is completed by noting that $E_R[M] = \sum_{i=1}^{n}\Phi(-\frac{\sqrt{k}|\rho_i|}{\xi_{\rho_i}})$ as $k \to \infty$, which could be easily derived from Central Limit Theorem and Proposition 1.

$\square$

## A.7 Proof of Theorem 6

*Proof.* By applying Lemma A1 we have

$$E_{Y|R}[L_Q(\hat{\beta}_Q^*)] - L_Q(\beta_Q^*) \le \gamma\frac{k}{n}. \tag{5}$$

Since $\beta_Q^*$ is the minimizer of the squared loss in the quantized space, by elementary algebra we have that

$$L_Q(\beta_Q^*) \le L_Q(\frac{1}{\sqrt{k}(1 - D_Q)}R^T\beta^*)$$
$$= \frac{1}{n}E_{Y|R}[\|Y - \frac{1}{k(1 - D_Q)}Q(XR)R^T\beta^*]$$
$$\overset{(a)}{=} \frac{1}{n}E_{Y|R}[\|Y - X\beta^*\|^2] + \frac{1}{n}\|X\beta^* - \frac{1}{k(1 - D_Q)}Q(XR)R^T\beta^*\|^2$$
$$= L(\beta^*) + (\beta^*)^T\Sigma\beta^* - \frac{2}{nk(1 - D_Q)}(\beta^*)^T RQ(XR)^T X\beta^*$$
$$+ \frac{1}{nk^2(1 - D_Q)^2}(\beta^*)^T RQ(XR)^T Q(XR)R^T\beta^*, \tag{6}$$

where $(a)$ is due to $Y - X\beta^* = \epsilon$ is $i.i.d$ zero-mean Gaussian independent of $R$. Here, the factor $\frac{1}{1 - D_Q}$ is again related to cosine estimation, and we will provide some discussions at the end of the proof. Recall the notation $X = [x_1, ..., x_n]^T$ with $x_i$ having unit norm, and $R = [r_1, ..., r_k]$.

We denote $z_{ip} \triangleq \langle x_i, r_p \rangle$. Hence, the quantized matrix $Q(XR)$ has $z_{ip}$ as the $(i,p)$-th entry, for $i = 1, ..., n$ and $p = 1, ..., k$. It is obvious that

$$\begin{pmatrix} z_{ip} \\ z_{jp} \end{pmatrix} \sim N\left( \begin{pmatrix} 0 \\ 0 \end{pmatrix}, \begin{pmatrix} 1 & \rho_{ij} \\ \rho_{ij} & 1 \end{pmatrix} \right), \tag{7}$$

where $\rho_{ij} = \langle x_i, x_j \rangle$. Moreover, Lemma A3 then gives $E[z_{ip}Q(z_{jp})] = (1 - D_Q)\langle x_i, x_j \rangle$. Further denote $\tilde{\beta} = \beta^* / \|\beta^*\|$ the standardized true parameter vector. It follows that

$$\begin{aligned} E[(\beta^*)^T RQ(XR)^T X\beta^*] &= E[(\beta^*)^T RQ(XR)^T] X\beta^* \\ &= E\left[ (\tilde{\beta})^T RQ(XR)^T \right] X\beta^* \|\beta^*\| \\ &= E\left[ \left[ \sum_{p=1}^k z_{\tilde{\beta},p} Q(z_{1p}), ..., \sum_{p=1}^k z_{\tilde{\beta},p} Q(z_{np}) \right]^T \right] X\beta^* \|\beta^*\| \\ &\overset{(b)}{=} k(1 - D_Q)(\beta^*)^T X^T X\beta^* \\ &= nk(1 - D_Q)(\beta^*)^T \Sigma \beta^*. \end{aligned} \tag{8}$$

Here, $z_{\tilde{\beta},p} = \langle \tilde{\beta}, r_p \rangle$, and $(b)$ is due to Lemma A3. Note that for $(x, y)$ following distribution (7) with cosine $\rho$, we have

$$\begin{aligned} E[x^2 Q(y)^2] &= E[(\rho y + \sqrt{1 - \rho^2} W)^2 Q(y)^2] \\ &= \rho^2 \xi_{2,2} + (1 - \rho^2)(1 - D_Q), \end{aligned} \tag{9}$$

where $W \sim N(0,1)$ is independent of $x, y$, and $\xi_{2,2} \triangleq E[y^2 Q(y)^2]$ for $y \sim N(0,1)$. Denote $\rho_{\tilde{\beta},i} = \langle \tilde{\beta}, x_i \rangle$. Now we can obtain

$$\begin{aligned} &E[(\beta^*)^T RQ(XR)^T Q(XR) R^T \beta^*] \\ &= \|\beta^*\|^2 E\left[ \tilde{\beta}^T RQ(XR)^T Q(XR) R^T \tilde{\beta} \right] \\ &= \|\beta^*\|^2 E\left[ \sum_{i=1}^n \left( \sum_{p=1}^k z_{\tilde{\beta},p} Q(z_{ip}) \right)^2 \right] \\ &= \|\beta^*\|^2 \sum_{i=1}^n E\left[ \sum_{p=1}^k z_{\tilde{\beta},p}^2 Q(z_{ip})^2 + \sum_{p=1}^k \sum_{q \neq p}^k z_{\tilde{\beta},p} Q(z_{ip}) z_{\tilde{\beta},q} Q(z_{iq}) \right] \\ &= \|\beta^*\|^2 \sum_{i=1}^n \left[ k(\xi_{2,2}\rho_{\tilde{\beta},i}^2 + (1 - \rho_{\tilde{\beta},i}^2)(1 - D_Q)) + k(k-1)(1 - D_Q)^2 \rho_{\tilde{\beta},i}^2 \right]. \end{aligned} \tag{10}$$

In the above, (10) holds because of (9) and the fact that $z_{\tilde{\beta},p} Q(z_{ip})$ is independent of $z_{\tilde{\beta},q} Q(z_{iq})$ for any $p \neq q$. By noticing that

$$\sum_{i=1}^n \rho_{\tilde{\beta},i}^2 = \frac{(\beta^*)^T X^T X\beta^*}{\|\beta^*\|^2} = \frac{n(\beta^*)^T \Sigma \beta^*}{\|\beta^*\|^2},$$

we can further have

$$\begin{aligned} &E[(\beta^*)^T RQ(XR)^T Q(XR) R^T \beta^*] \\ &= \|\beta^*\|^2 \left[ k(\xi_{2,2} \sum_{i=1}^n \rho_{\tilde{\beta},i}^2 + (n - \sum_{i=1}^n \rho_{\tilde{\beta},i}^2)(1 - D_Q)) + k(k-1)(1 - D_Q)^2 \sum_{i=1}^n \rho_{\tilde{\beta},i}^2 \right] \\ &= nk(1 - D_Q)\|\beta^*\|^2 + n\left[ k(\xi_{2,2} - 1 + D_Q) + k(k-1)(1 - D_Q)^2 \right] (\beta^*)^T \Sigma \beta^*. \end{aligned} \tag{11}$$

Now, taking expectation on both sides of (6) $w.r.t.$ $R$ and combining (8) and (11), we have

$$\begin{aligned} &E_R[L_Q(\beta_Q^*)] \\ &\leq L(\beta^*) + \left[ 1 - 2 + \frac{\xi_{2,2} - 1 + D_Q}{k(1 - D_Q)^2} + \frac{k-1}{k} \right] (\beta^*)^T \Sigma \beta^* + \frac{1}{k(1 - D_Q)}\|\beta^*\|^2 \\ &= L(\beta^*) + \frac{1}{k}\|\beta^*\|_\Omega^2, \end{aligned} \tag{12}$$

where $\Omega = [\frac{\xi_{2,2}-1+D_Q}{(1-D_Q)^2} - 1]\Sigma + \frac{1}{1-D_Q}I_d$, with $\|\beta^*\|_\Omega = \sqrt{(\beta^*)^T\Omega\beta^*}$, and $I_d$ the identity matrix. Lastly, taking expectation $w.r.t.$ R in (5), we obtain

$$E_{Y,R}[L_Q(\hat{\beta}_Q^*)] \leq E[L_Q(\beta_Q^*)] + \gamma\frac{k}{n}$$
$$\leq \gamma\frac{k}{n} + L(\beta^*) + \frac{1}{k}\|\beta^*\|_\Omega^2.$$

This completes the proof. Now we briefly discuss the role of factor $\frac{1}{1-D_Q}$ in (6). Note that in our model, $X\beta^* = \|\beta^*\|X\tilde{\beta} = \|\beta^*\|[\rho_{\tilde{\beta},1},...,\rho_{\tilde{\beta},n}]^T$ can be regarded as the scaled cosine between data vectors and the true parameter, and $Q(XR)R^T\beta^*$ is then a biased estimator of $X\beta^*$ with mean equal to $(1 - D_Q)X\beta^*$, according to Lemma A3. Therefore, the factor $\frac{1}{1-D_Q}$ acts as a debiasing operation—Similar in spirit to the previous analysis for classification problems. $\square$