[Reviews · NeurIPS 2019]

Reviewer 1



This paper presents solid theoretical work that is well written in clear, at least to the extent that is possible for a dense topic. It address relevant questions about generalization error for random projection combined with quantization and provides useful insights after each results. The choice of simple but meaningful models for which a broad set of people are likely to have decent intuition, also makes the work more accessible. The paper is not without its shortcomings however. Most of which I cover in section 5 below. That being said I struggle with one of the key assumptions in the paper which is the fact that we can normalize all the data to the unit circle. The author(s) justification is that it is a “standard preprocessing step for many learning algorithms”. This certainly has not been my personal experience, as opposed to say mean/variance or min/max normalization. For high dimensional data I can conceive this transformation to be a reasonable choice, but I definitely feel it requires more than a passing mention. It either needs established references or a separate analysis, potentially empirical, of how this affects performance.

Reviewer 2



Overall it is an interesting paper, with nice results. However, I am not as certain about the relevance of the results. It is not clear that the three methods considered are often used in their compressive version combined with quantization. One example I have seen is: [*] M. Li, S. Rane, and P. T. Boufounos, “Quantized embeddings of scale-invariant image features for mobile augmented reality,” IEEE 14th International Workshop on Multimedia Signal Processing (MMSP), Banff, Canada, Sept. 17-19, 2012, and subsequent work by the authors above. Still, the literature is still scarce for such applications. It would be interesting if the authors could provide a bit more motivation. With respect to the results, I think that overall it is a good start in this direction. One thing that is missing from the results is the link between the quantization interval \Delta and the number of best used. I can always reduce \Delta and improve the error bounds, but then I need to use more bits per measurement. Furthermore, when it comes to compressive methods, there is also the tradeoff between number of measurements and size of \Delta, which affects the total rate. Some discussion towards addressing these questions would be valuable. ==== I've seen and taken into account the author's response, and it does not change my score.

Reviewer 3



In general, I think the quality of this paper is good and it is written in a clear form. I think the content of this paper meet the quality of the conference. However, I have a concern about the title of the paper. Is it appropriate to use the term of "generalization error". From my understanding, generalization error refers to the difference of error on testing set and training set, given the same parameters. The error bound in Theorem 5 exactly shows the generalization error. However, I'm not sure whether the other theorems refers to generalization error. (1) In Theorem 2 and 3 , lower case (x,y) refers to test data. LHS is testing error with expectation over (x,y). The first term in RHS does not contain expectation over (x,y), so this is strange. I think it should be just L(h*), please check. If it were L(h^*), it is Bayer risk, not training error. So I think Theorem 2 and 3 can not be called "generalization" error. They should be called "compression error" and "quantization and compression error". (2) In Theorem 6, similarly both terms in the LHS refers to errors on test data, so it should not be "generalization error". Also, L_Q(\hat{\beta}*_Q) is already expected over Y|R so there is no need to take expectation over Y and R. I'm not blaming the significance of these bounds, but I think there should be a more principled way to make the connections over there three bounds, rather than using the term "generalization error". Minor suggestion: I wish to see more discussions about previous work on quantized compressive learning. The authors provide some references in the introduction, but I wish to see more detailed comparisons of results of the previous work and the current work. This will make the main contribution of this work more clear. ============================================================== I'm happy that the authors have answered my question well by providing a good reference. I also learned something new from their feedback. I would like to increase my score from 6 to 7.

[Author Response · NeurIPS 2019]

1. **Response to Reviewer 1**: We appreciate your valuable & insightful comments and suggestions.

2. *Data normalization.* Yes, it is an issue which is worth more discussion. In this paper, we assumed that all data instances
3. have unit $l_2$ norm. We adopted this assumption partly due to a seemly common observation that instance normalization
4. often improves classification/regression/clustering results. For example, if one examines the datasets in the LIBSVM
5. (cited as [5]) website, for most of the datasets, all instances are normalized to have unit $l_2$ norm. There are also similar
6. observations in the deep learning literature; see for example, the paper by Ulyanov et.al.: *Instance Normalization: The*
7. *Missing Ingredient for Fast Stylization*, arXiv:1607.08022.

8. Even if the instance norms are not 1, one can often assume they are known
9. because that only requires storing one real number per instance. With known
10. norms, LM quantization is essentially the same, that is, we quantize data by
11. scaling the quantizer according to the norm of each vector. In some application
12. (e.g., regression), assuming instance normalization simplified the analysis as one
13. does not have to keep track of the norms in the calculations. We will expand the
14. discussions on the impact of the assumption of unit norm. Thank you.

15. *Debiased variance.* Yes, this is an interesting problem and we believe it should
16. be possible to design quantizers that aim at reducing debiased variance. And yes,
17. this is a meaningful and interesting topic to study. Thanks for pointing this out.

18. *Additional experiments on regression.* Thanks for the nice suggestion. Figure 1
19. is a simulated result of OLS. We report test mean squared error (MSE) of fitting
20. OLS using different strategies. For uniform quantizer, we set the largest finite
21. boarders equal to corresponding LM quantizer to make fair comparison. LM
22. outperforms uniform quantization on this task. More results of this kind can be
23. reported in the supplementary material, as you kindly suggested.

Figure 1: Simulation on OLS problem. $n = 3000$, $d = 1200$. Both $X$ and $\beta$ are generated from iid $N(0,1)$ and $X$ is normalized. Gaussian noise variance $\sigma^2 = 0.2$.

24. **Response to Reviewer 2**: Thanks for your valuable comments and suggestions. We would like to elaborate on the
25. motivation of "compressed + quantized" learning. It should be now clear that "compressed learning" is a popular
26. topic in the past 10 years, with many good papers in premier conference proceedings and journals, for a wide range
27. of applications: similarity search, clustering, classification, regression, etc. Because one will have to store/transmit
28. the compressed data and use them for subsequent calculations, it is a natural step to consider quantized version of
29. compressed learning. Besides the papers written by Boufounos and collaborators, there is already a fairly rich literature
30. on quantized random projections, often in non-machine-learning venues, for example,

31. *Quantized Compressive Sensing*, by Zymnis, Boyd and Candès, IEEE Signal Process. Lett., 2010; and
32. *Robust 1-bit compressed sensing and sparse logistic regression: A convex programming approach* by Plan and Vershynin,
33. IEEE Trans. Information Theory, 2013, among other papers written by prominent researchers.

34. Before this submission, theoretical analysis, especially on learning performance using quantized compressive data, has
35. not been conducted yet. In recent years, as data size becomes larger and larger, data compression is becoming more and
36. more important. Thus, we hope our work will be useful both theoretically and practically.

37. Also, thanks for suggesting to exploit the trade-off between number of bits, number of projections, and accuracy. Since
38. the derived bounds are functions of these parameters, we agree it is beneficial to generate plots to show the trade-off.

39. **Response to Reviewer 3**: Thanks so much for raising the interesting and very important issue regarding the definition
40. of "generalization error". We agree the definitions in the literature are not always consistent. You are correct that
41. Theorem 2 and Theorem 3 are about "Bayes Risk". Interestingly, "Bayes risk" in the context of near-neighbor classifiers
42. is sometimes also referred to as "generalization error"; see reference [25], the well-known textbook in machine learning:

43. *Understanding Machine Learning: From Theory to Algorithms*, by Shai Shalev-Shwartz and Shai Ben-David. 2014

44. Chapter 19.2.1, entitled "*A Generalization Bound for the 1-NN Rule*" , derives the "generalization bounds" for 1-NN
45. classifiers, where the RHS terms are indeed "Bayes Risk". The book is available online. While we are not allowed
46. to provide links here, it is fairly easy to find. We certainly do not mean to "blame on" this nice book. We agree with
47. Reviewer 3 that one should be more consistent with the definitions. We will think carefully what might be a more
48. suitable title. Perhaps simply removing "Generalizing" from the current title might be an option?

49. The issue is similar for regression. There are quite a few papers which called regression test error as "generalization
50. error", e.g., see *Compressed Least-Squares Regression*, by Odalric-Ambrym Maillard and Rémi Munos, NIPS 2009.

51. Again, we do not blame on prior papers for the inconsistency regarding definitions. We will think about this issue
52. carefully and might also consult other experts. Thanks also for other suggestions on improving the quality of the work.

[Meta-Review · NeurIPS 2019]

This is a strong and well written theoretical paper. Certainly one of the three best in my stack. I strongly recommend acceptance as a Spotlight presentation. [This meta-review was reviewed and revised by the Program Chairs]